# USP4 promotes the proliferation and glucose metabolism of gastric cancer cells by upregulating PKM2

**Yuanyuan Chen[1]◉, Yunfei Guo[1]◉, Mei Yuan[2], Song Guo[1], Shuaishuai Cui[1], Dahu Chen[1]\***

1 School of Life Sciences and Medicine, Shandong University of Technology, Zibo, China, 2 School of Life Sciences, Jiangsu Normal University, Xuzhou, China

◉ These authors contributed equally to this work.
\* dahuchen@sdut.edu.cn

**Data Availability Statement:** All relevant data are within the paper and its Supporting Information files.

**Funding:** This work was supported by National Natural Science Foundation of China (81872005) to

## Abstract

### Background

The pyruvate kinase enzyme PKM2 catalyzes the final step in glycolysis and converts phosphoenolpyruvate (PEP) to pyruvate. PKM2 is often overexpressed in cancer and plays a role in the Warburg effect. The expression of PKM2 can be regulated at different levels. While it has been proven that PKM2 can be regulated by ubiquitination, little is known about its de-ubiquitination regulation.

### Methods

Immunoprecipitation was applied to identify the PKM2 interaction protein and to determine the interaction region between PKM2 and USP4. Immunofluorescence was performed to determine the cellular localization of USP4 and PKM2. The regulation of PKM2 by USP4 was examined by western blot and ubiquitination assay. MTT assays, glucose uptake, and lactate production were performed to analyze the biological effects of USP4 in gastric cancer cells.

### Results

USP4 interacts with PKM2 and catalyzes the de-ubiquitination of PKM2. Overexpression of USP4 promotes cell proliferation, glucose uptake, and lactate production in gastric cancer cells. Knockdown of USP4 reduces PKM2 levels and results in a reduction in cell proliferation and the glycolysis rate.

### Conclusions

USP4 plays a tumor-promoting role in gastric cancer cells by regulating PKM2.

DC. The sponsor plays no role in the study design, data collection and analysis, decision to publish, or preparation of the manuscript.

**Competing interests:** The authors have declared that no competing interests exist.

## Introduction

Gastric cancer is a malignant tumor that develops in the stomach lining. It is the fifth most common cancer and the third leading cause of cancer-related deaths worldwide [1]. Gastric cancer is rarely diagnosed at an early stage; therefore, it is associated with high mortality. Despite the improvements in surgical techniques, advances in chemotherapy and radiation therapy, and newly developed targeted therapy, the 5-year overall survival rate of gastric cancer patients is still lower than 25% [2]. Therefore, there is an urgent need to understand the molecular mechanisms of gastric cancer progression and find novel, promising therapy targets.

Growing evidence suggests that metabolic aberrations are closely linked to cancer development and progression. One of the most important characteristics of solid tumors is their energy metabolism reprogramming. In this process, cancer cells alter their metabolism to support their growth and survival. In normal cells, glucose is primarily metabolized through oxidative phosphorylation in the mitochondria to produce ATP. However, cancer cells preferentially use glycolysis to generate energy even in the presence of oxygen, a phenomenon known as the Warburg effect [3]. One of the key changes in the reprogramming of energy metabolism in cancer cells is an increase in glucose uptake and its conversion to lactate by the enzyme lactate dehydrogenase A (LDHA). The Warburg effect is a phenotypic characteristic of cancer metabolism and a key factor for tumor cell proliferation and metastasis [4, 5].

Pyruvate kinase is a key enzyme that catalyzes the final step of glycolysis, the process by which glucose is broken down to generate ATP. Four isoforms of pyruvate kinase were found. The L (liver) and R (red blood cells) isoforms are expressed in specific tissues, while the M1 (muscle) and M2 (fetal) isoforms are expressed in most tissues, including cancer cells [6]. Pyruvate kinase M2 (PKM2) has gained considerable attention in cancer research due to its role in promoting the Warburg effect. As the final rate-limiting enzyme of cell glycolysis, PKM2 plays a critical role in tumor cell metabolism reprogramming from oxidative phosphorylation to aerobic glycolysis [7, 8] and is essential for aerobic glycolysis in tumors (Warburg effect) [9, 10]. A large number of studies have shown a correlation between PKM2 expression and solid tumors of the digestive system. PKM2 was found to be overexpressed in gastric tumor tissues compared to normal tissues, and its expression level was associated with poor survival in gastric cancer patients [11–13]. PKM2 promotes gastric cancer cell migration and inhibits cell autophagy through activating the PI3K/AKT signaling pathway, which contributes to the malignant development of gastric cancer [14]. These findings indicate that PKM2 expression plays an essential role in gastric cancer, and there is a need to further understand the mechanisms regulating PKM2 expression during tumor development.

Gene expression is regulated at various levels, such as transcription, mRNA processing, translation, and post-translational modifications. Post-translational modifications can also influence protein expression and function. For instance, phosphorylation can change protein activity or localization, while ubiquitination can direct proteins to degradation by the proteasome. Ubiquitination involves the covalent ligation of the small protein ubiquitin to lysine residues on target proteins, which can induce protein degradation and alter protein localization [15]. The ubiquitination process is catalyzed by three types of enzymes: E1 (ubiquitin-activating enzymes), E2 (ubiquitin-conjugating enzymes), and E3 (ubiquitin ligases) [16] and can be offset by its reverse process, de-ubiquitination, where deubiquitinating enzymes (DUBs) remove ubiquitin from the target protein [17]. Thus far, the regulation of PKM2 by ubiquitination or deubiquitination modification has been documented. For example, Parkin is an identified PKM2 E3 ligase that interacts with PKM2 to regulate its activity instead of its stability [18]. The Laforin/Malin complex can polyubiquitinate PKM2 and impair its nuclear localization [19]. Additionally, Baek et al. found that USP20 interacts with PKM2 and downregulates

the ubiquitination level of PKM2 [20]. However, research on the ubiquitination and de-ubiquitination modifications of PKM2 is limited.

Ubiquitin-specific proteases (USPs) are the largest family of DUBs, with over 60 members in humans [21]. USP4 is a member of the USP family that can function as either a tumor promoter or a tumor suppressor, depending on cancer type. USP4 has been associated with many human malignant tumors, including brain cancer [22], liver cancer [23, 24], colorectal cancer [25, 26], and melanoma [27]. On the other hand, USP4 can establish its tumor suppressive role by inhibiting cell proliferation or promoting cell apoptosis in neck squamous cell carcinoma [28] and breast cancer [29]. These reports suggested that the impact of USP4 on cancer biology is tumor type-specific and tissue context-dependent. Nevertheless, the relevant pathogenic roles of USP4 in gastric cancer have not been well investigated and require further exploration.

In this study, we found that USP4 interacts with PKM2 and catalyzes PKM2 de-ubiquitination. USP4 affects cell proliferation and enhances the Warburg effect in gastric cancer cells by upregulating PKM2.

## Materials and methods

### Cell culture and proliferation

AGS, HGC-27, and HEK-293T were maintained either in RPMI 1640 medium or DMEM (Gibco, NY) supplemented with 10% fetal bovine serum and 1% penicillin-streptomycin within a humidified atmosphere containing 5% $CO_2$ at 37˚C.

Cell proliferation was determined by the MTT assay and the colony formation assay. The MTT assay was performed according to the instructions of the manufacturer. For the colony formation assay, 10,000 cells were seeded into a 10-cm dish and cultured for 15 days. The formed colonies were fixed, stained, and counted.

### Plasmids and transfection

Human USP4 ORF and PKM2 ORF were amplified by PCR and subcloned into Flag-tagged vectors and Myc-tagged vectors, respectively. A catalytically inactive form of USP4 (C311A) was constructed by site-directed mutagenesis. For the deletion mutant constructs of either USP4 or PKM2 (S1 and S2 Tables), the fragments amplified by PCR were cloned into Flag-tagged or Myc-tagged vectors. USP4 shRNAs and PKM2 shRNAs were purchased from Genecopoeia. DNA constructs and shRNA clones were transiently or stably transfected into cells with Lipofectamine 3000. The relevant antibiotics were used to screen the stable cell clones.

### Immunoprecipitation

HEK-293T cells were transfected with the corresponding plasmids. Cells were harvested and lysed. The whole cell lysates were collected for immunoprecipitation with the relevant antibodies. Normal mouse or rabbit serum was used as a negative control. For detecting the endogenous USP4 and PKM2 interaction, the cell lysates were incubated with the USP4 antibody. The precipitated immunocomplexes by protein A/G PLUS-Agarose beads were collected, analyzed by SDS-PAGE, and immunoblotted with relevant antibodies.

### Immunobloting

Western blot analysis was performed using standard methods and detected by enhanced chemiluminescence. The following antibodies from Proteintech were used: anti-GAPDH antibody (60004-1-lg), anti-Myc antibody (16286-1AP), anti-Flag antibody (66008-4-lg), anti-HA

antibody (51064-2-AP), anti-USP4 antibody (66822-1-lg), and anti-PKM2 antibody (15822-1-AP).

## Quantitative RT-PCR

Total RNA was extracted, and the cDNA samples were synthesized. The prepared cDNA samples were analyzed by quantitative real-time RT-PCR with Power SYBR Green PCR Master Mix (Bio-Rad).

Primer sequences were as follows:

Glut1 forward, 5′ –GGCCAAGAGTGTGCTAAAGAA– 3′;

Glut1 reverse, 5′ –ACAGCGTTGATGCCAGACAG– 3′;

LDHA forward, 5′ –ATGGCAACTCTAAAGGATCAGC– 3′;

LDHA reverse, 5′ –CCAACCCCAACAACTGTAATCT–3′;

18S RNA forward, 5′–CTACCACATCCAAGGAAGCA– 3′;

18S RNA reverse, 5′ –TTTTTCGTCACTACCTCCCCG–3′

Each real-time PCR reaction was conducted in triplicate. The comparative CT method (2 CT) was used for the calculation of relative gene expression.

## In vivo ubiquitination assay

Flag-USP4 or Flag-USP4 (C311A), HA-ubiquitin, and Myc-PKM2 were co-transfected into HEK-293T cells. After transfection for 42 hours, cells were treated with 5 M MG132 for 4 hours and then harvested. The harvested cells were lysed and incubated with Myc antibodies and then with Protein G sepharose beads. The immunoprecipitates were subjected to western blotting analysis. PKM2 proteins were immunoprecipitated by Myc antibodies, and ubiquitinated PKM2 was detected by anti-HA antibodies.

## Immunofluorescence

HEK-293T cells were cultured on coverslips and co-transfected with Flag-USP4 and Myc-PKM2. After transfection, cells were washed and fixed with 4% paraformaldehyde. Next, cells were blocked with 10% nonfat milk and probed with rabbit anti-Flag and mouse anti-Myc antibodies at 4˚C overnight. Then, the cells were washed and incubated with a 1/200 dilution of anti-mouse Texas Red and anti-rabbit Alexa Fluor 488. The coverslips were mounted onto glass slides using a mounting solution with DAPI.

## Measurements of glucose metabolism

The glucose uptake and lactate production were measured by using the glucose uptake colorimetric assay kits (K676-100) and lactate colorimetric assay kits (K627-100) from BioVision.

## Statistical analysis

The data were presented as the means and standard deviation of three independent experiments. Differences between groups were analyzed using the Student's *t-test*, and the differences were considered statistically significant at *, $P < 0.05$ and **, $P < 0.01$.

# Results

## USP4 is a PKM2-interacting protein

PKM2 can be regulated by ubiquitination [18, 19] and de-ubiquitination [20]. De-ubiquitination is a reverse process of ubiquitination where the ubiquitin can be removed with the help of

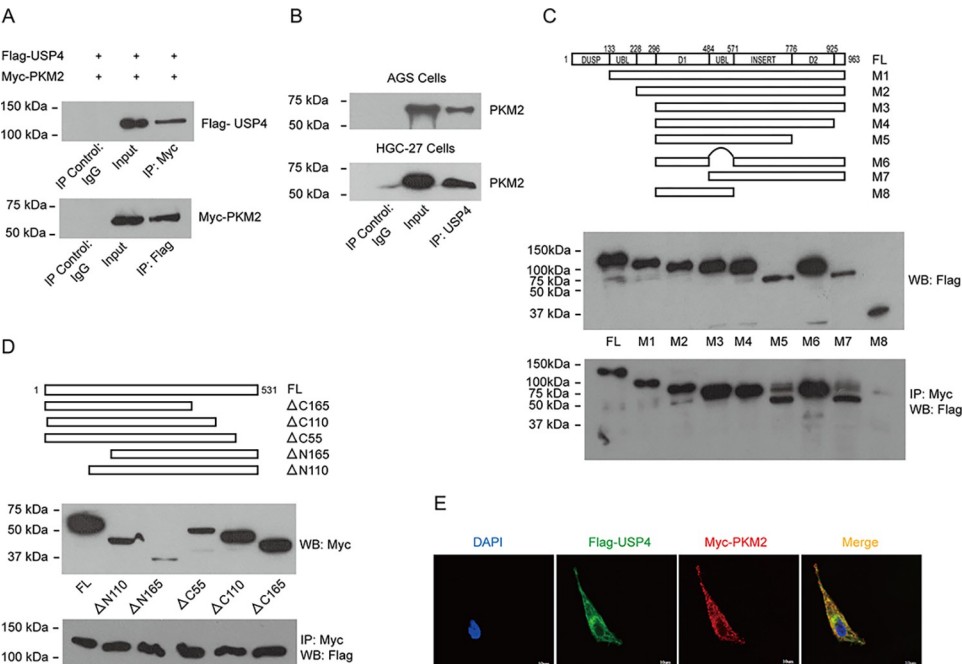

**Fig 1. USP4 interacts with PKM2.** (A) Top: 293T cells were transfected with Flag-USP4 and Myc-PKM2, immunoprecipitated with Myc antibody, and immunoblotted with Flag antibody. Bottom: 293T cells were transfected with Flag-USP4 and Myc-PKM2, immunoprecipitated with Flag antibody, and immunoblotted with Myc antibody. (B) Endogenous PKM2 was immunoprecipitated from either AGS cells (top) or HGC-27 cells (bottom) with the USP4 antibody and immunoblotted with the PKM2 antibody. (C) Top: Schematic representation of flag-tagged full-length USP4 (FL) and its various deletion mutants (M1–M8). Middle: The expression of flag-tagged full-length USP4 (FL) and its various deletion mutants (Input). Bottom: Flag-USP4 and its eight deletion mutants were co-transfected into 293T cells with Myc-tagged PKM2, respectively, immunoprecipitated with Myc antibody, and immunoblotted with Flag antibody. (D) Top: Schematic representation of Myc-tagged full-length PKM2 (FL) and its five deletion mutants. Middle: The expression of Myc-tagged PKM2 and its five deletion mutants (input). Bottom: Myc-PKM2 and its five deletion mutants were co-transfected into 293T cells with Flag-tagged USP4, respectively, immunoprecipitated with Myc antibody, and immunoblotted with Flag antibody. (E) Immunofluorescent staining of USP4 (green) and PKM2 (red) in 293T cells transfected with Flag-USP4 and Myc-PKM2. The left panel is nuclear 4',6-diamidino-2-phenylindole (DAPI; blue). The right panel is the overlay of USP4 (green), PKM2 (red), and DAPI (blue) staining of the same field. Scale bar: 10 m.

deubiquitinating enzymes (DUBs) [17, 30, 31]. Ubiquitin-specific protease is one of the DUBs that regulate ubiquitin-dependent pathways by cleaving ubiquitin-protein bonds [32]. We postulated that some USPs may be involved in the regulation of PKM2.

In order to investigate the putative deubiquitylases for PKM2, we constructed a 20-ubiquitin-specific protease miniexpression library (with a flag-tag). These USPs were co-transfected with Myc-tagged PKM2 into HEK-293T cells. Immunoprecipitation assays showed that USP4 was detected in Myc antibody immunoprecipitated samples. To further verify the association between PKM2 and USP4, we performed co-immunoprecipitation experiments. We found that the anti-Myc antibody co-precipitated Flag-USP4, whereas the control serum did not (Fig 1A). Similarly, anti-Flag antibodies co-precipitated Myc-PKM2 (Fig 1A). Furthermore, we sought to determine whether USP4 interacts with PKM2 in vivo. We found that PKM2 was detected in the anti-USP4 immunoprecipitates but not in the normal rabbit IgG immunoprecipitates in AGS and HGC-27 cells (Fig 1B).

To determine the region of USP4 involved in the interaction with PKM2, eight deletion mutants of USP4 were constructed (Fig 1C). We found that the insert region of USP4 (amino acid residues 571 to 776) is required for interacting with PKM2. To characterize the minimal

region of PKM2 required for binding USP4, we co-transfected five different PKM2 deletion mutants and USP4 into 293T cells and performed immunoprecipitations. We found that all five deletion mutants can interact with USP4 (Fig 1D). Although we are not able to determine the exact region of PKM2 that interacts with USP4 based on our current results, we postulate that the interacting region of PKM2 with USP4 may localize in the overlapping region of the five deletion mutants. This needs to be verified through further investigation.

We next performed an immunofluorescence assay to check the localization of these two proteins. We found that both USP4 and PKM2 were co-localized in the cytosol and nucleus (Fig 1E). Taken together, our biochemical assay revealed that USP4 was specifically bound to PKM2.

## USP4 upregulates PKM2 by promoting PKM2 de-ubiquitination

The above results prove that USP4 interacts with PKM2. Because USP4 contains a carboxyl-terminal ubiquitin hydrolase domain and is a peptidase, we sought to determine whether USP4 regulates PKM2 as a novel PKM2 deubiquitylase by catalyzing its de-ubiquitination. We first investigated whether the expression of PKM2 is regulated by USP4. There is detectable basal expression of PKM2 in both ASS cells (Fig 2A, middle panel, left) and HGC-27 cells (Fig 2B, middle panel, left). We found that PKM2 expression levels are increased in USP4 over-expression cells (Fig 2A). However, the catalytically inactive form of USP4 (C311A) that still bears the capability of interacting with PKM2 (data not shown) has no effect on the expression of PKM2 (Fig 2A). These results indicated that USP4 can upregulate PKM2, and the regulatory

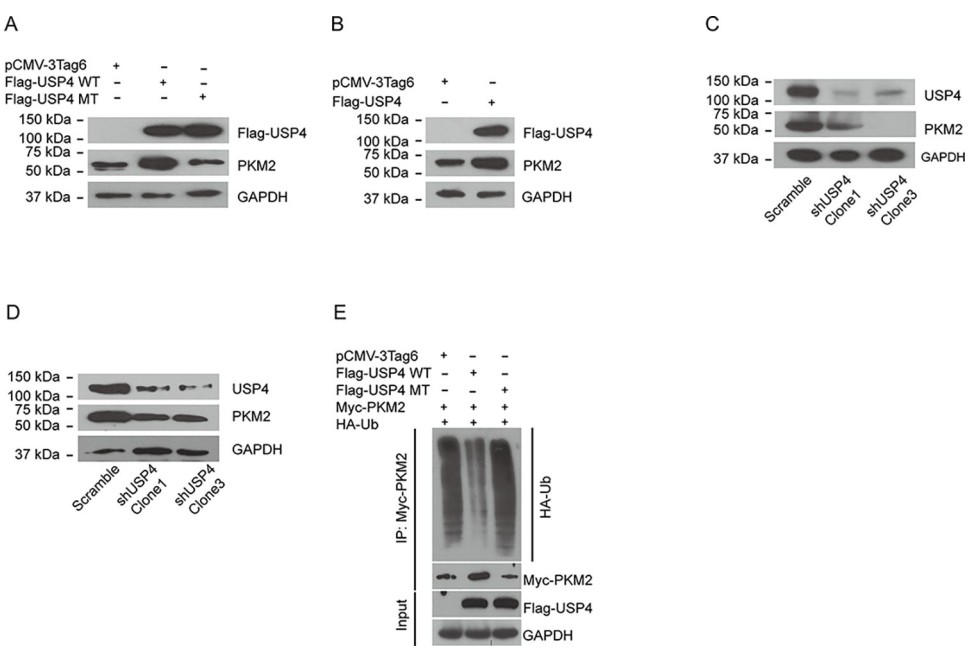

**Fig 2. USP4 upregulates PKM2 by preventing its degradation.** (A) Immunoblotting of USP4 (Flag) and PKM2 in AGC cells that are stably transfected with control plasmids, Flag-USP4 and Flag-USP4 mutant, respectively. (B) Immunoblotting of USP4 (Flag) and PKM2 in HGC-27 cells that are stably transfected with control plasmid and Flag USP4, respectively. (C) Immunoblotting of USP4 and PKM2 in AGS cells that are stably transfected with scramble plasmid, USP4 shRNA (clone 1), and USP4 shRNA (clone 3), respectively. (D) Immunoblotting of USP4 and PKM2 in HGC-27 cells that are stably transfected with scramble plasmids, USP4 shRNA (clone 1) and USP4 shRNA (clone 3), respectively. (E) 293T cells were co-transfected with Myc-PKM2, HA-ubiquitin (Ub), and Flag-USP4 or the USP4^C311A mutant, immunoprecipitated with Myc beads, and immunoblotted with antibodies to HA and Myc. Cells were treated with MG132 (10 M) for 4 hours before harvest.

capability of USP4 is related to its enzyme activity. In HGC-27 cells, we observed the similar regulation ability of USP4 and PKM2 (Fig 2B). While the endogenous USP4 was knocked down, the PKM2 level was decreased (Fig 2C and 2D). This result further confirmed the regulation of USP4 on PKM2.

As a deubiquitylase, USP4 can catalyze the reverse process of ubiquitination and change the ubiquitination level of target proteins, thereby preventing their degradation. We already found that USP4 can increase PKM2 expression, and the upregulation of PKM2 by USP4 is associated with its enzymatic activity. Therefore, we wonder if USP4 can directly catalyze the de-ubiquitination of PKM2. We performed the in vitro ubiquitination experiment and found that wild-type USP4, not the USP4-C311A mutant, indeed inhibited PKM2 ubiquitination (Fig 2E). Collectively, all these results suggest that USP4 is a deubiquitylase for PKM2 and a critical regulator of PKM2 in cells.

## USP4 promotes gastric cancer cell proliferation

PKM2 promotes glycolysis, which often increases the tumor cell's proliferation. Because USP4 upregulates PKM2, we thought that USP4 might be able to change the cell's proliferation. We examined the effect of USP4 on cell proliferation by using the MTT assay and the colony formation assay. First, we checked the proliferation of 293T cells. While the proliferation of cells is significantly increased when USP4 is overexpressed, the overexpression of mutant USP4 (the catalytically inactive form of USP4) showed no effects on cell proliferation (Fig 3A). This indicated that the enzymatic activity of USP4 is associated with its capability to promote cell proliferation. Then we checked the effect of USP4 on AGS cell proliferation. As shown in Fig 3A, USP4 dramatically increased the proliferation of AGS cells. To confirm that this phenomenon is not related to cell type, we used HGC-27 cells to perform the same assay. We found that USP4 had the same effect on the proliferation of HGC-27 cells (Fig 3A). In agreement with this, depletion of USP4 in AGS cells (Fig 2C) obviously reduced cell proliferation (Fig 3B).

Our results have shown that the enzymatic activity of USP4 is not only related to its upregulation of PKM2 but also to its capability for promoting cell proliferation. We wonder if the regulation of USP4 on cell proliferation is mediated by PKM2. For this purpose, we knocked down PKM2 in USP4 overexpression cells (S1 Fig) and then examined the proliferation of these cells. Although PKM2 is important for cell proliferation, its loss or significant reduction is not necessarily lethal to most cell types. Cells may upregulate other isoforms like PKM1 or shift to more oxidative phosphorylation and less aerobic glycolysis, allowing them to produce enough ATP to survive. We found that the promoting effect of USP4 on cell proliferation was diminished due to the knockdown of PKM2 (Fig 3C). This result clearly indicates that PKM2 is a mediator for USP4 function in gastric cancer cells.

Similarly, the colony formation assay results (Fig 3D–3F) indicated that USP4 promotes gastric cancer proliferation by regulating PKM2.

## USP4 promotes glucose uptake and lactate production in gastric cancer cells

PKM2 plays a crucial role in the Warburg effect. Because USP4 has been shown to regulate PKM2 in gastric cancer cells, we wonder if USP4 will affect the Warburg effect. The most typical features of glucose metabolism in cancer cells are the alternation of glucose uptake and lactate production. Therefore, we examined the glucose uptake and lactate production of gastric cancer cells in the presence or absence of USP4. It has been reported that PKM2 upregulates the expression of glycolysis-related genes, for example, Glut1 and LDHA [33]. GLUT1 and

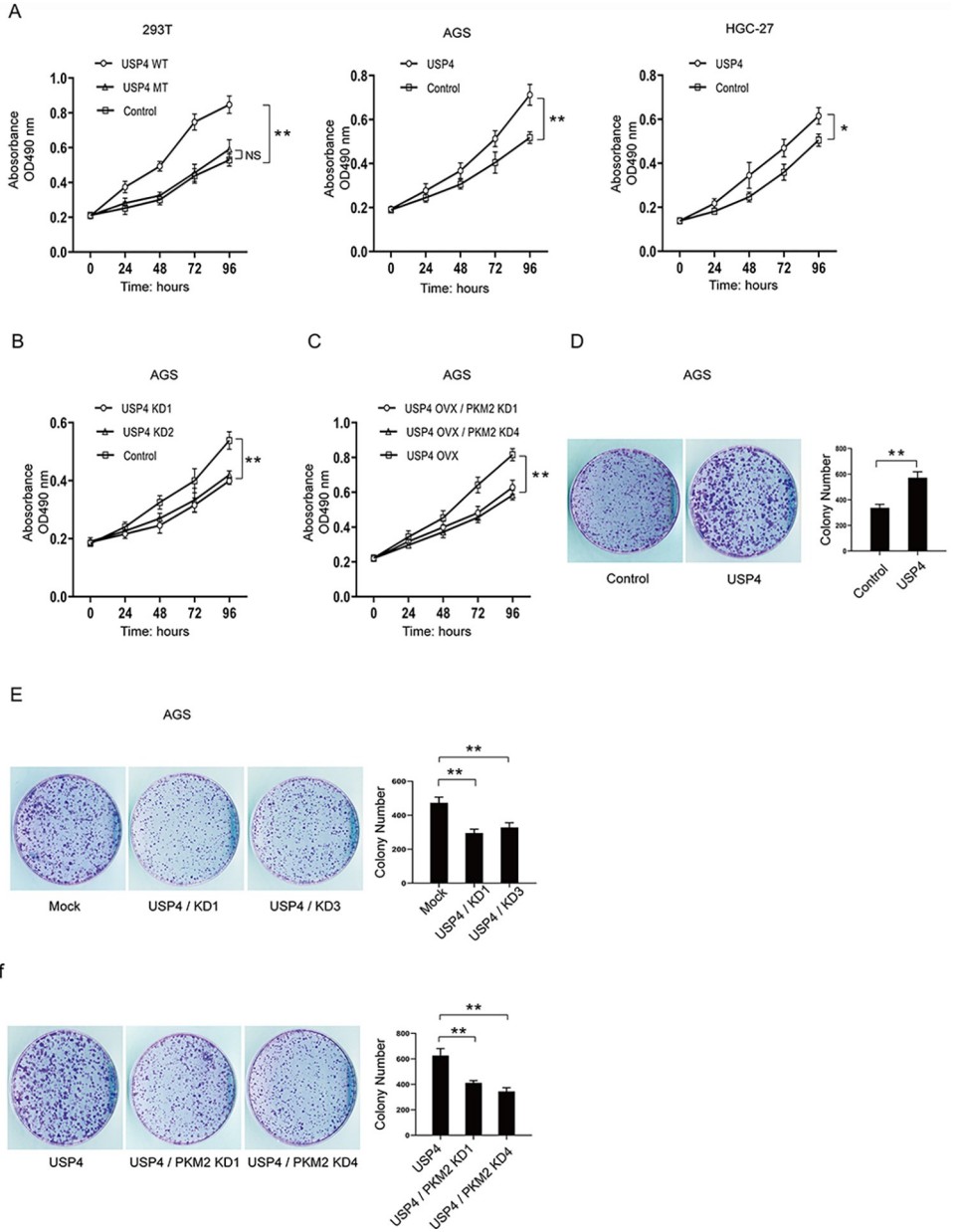

**Fig 3. USP4 promotes the proliferation of gastric cancer cells.** (A) Left: The cell proliferation of USP4- or USP4$^{C331A}$-transfected 293T cells Middle: The cell proliferation of AGS cells transfected with USP4. Right: The cell proliferation of HGC-27 cells transfected with USP4. (B) The cell proliferation of AGS cells transfected with USP4 shRNAs. (C) The cell proliferation of AGS cells transfected with USP4 and PKM2 shRNAs. (D) Representative images and statistical results of colony formation in AGS cells transfected with USP4. (E) Representative images and statistical results of colony formation in AGS cells transfected with USP4 shRNAs. (F) Representative images and statistical results of colony formation in AGS cells transfected with USP4 and PKM2 shRNAs.

LDHA are required for glucose uptake and the conversion of pyruvate to lactate, respectively [6]. Because USP4 upregulates PKM2 expression, we speculate that USP4 may increase these gene expressions due to its upregulation of PKM2. Thereby, we checked these genes expression by RT-qPCR. As shown in Fig 4A, the expression of both GLUT1 and LDHA was elevated when USP4 was overexpressed. On the other hand, knockdown of USP4 in gastric cancer cells led to a decrease in the expression of GLUT1 and LDHA.

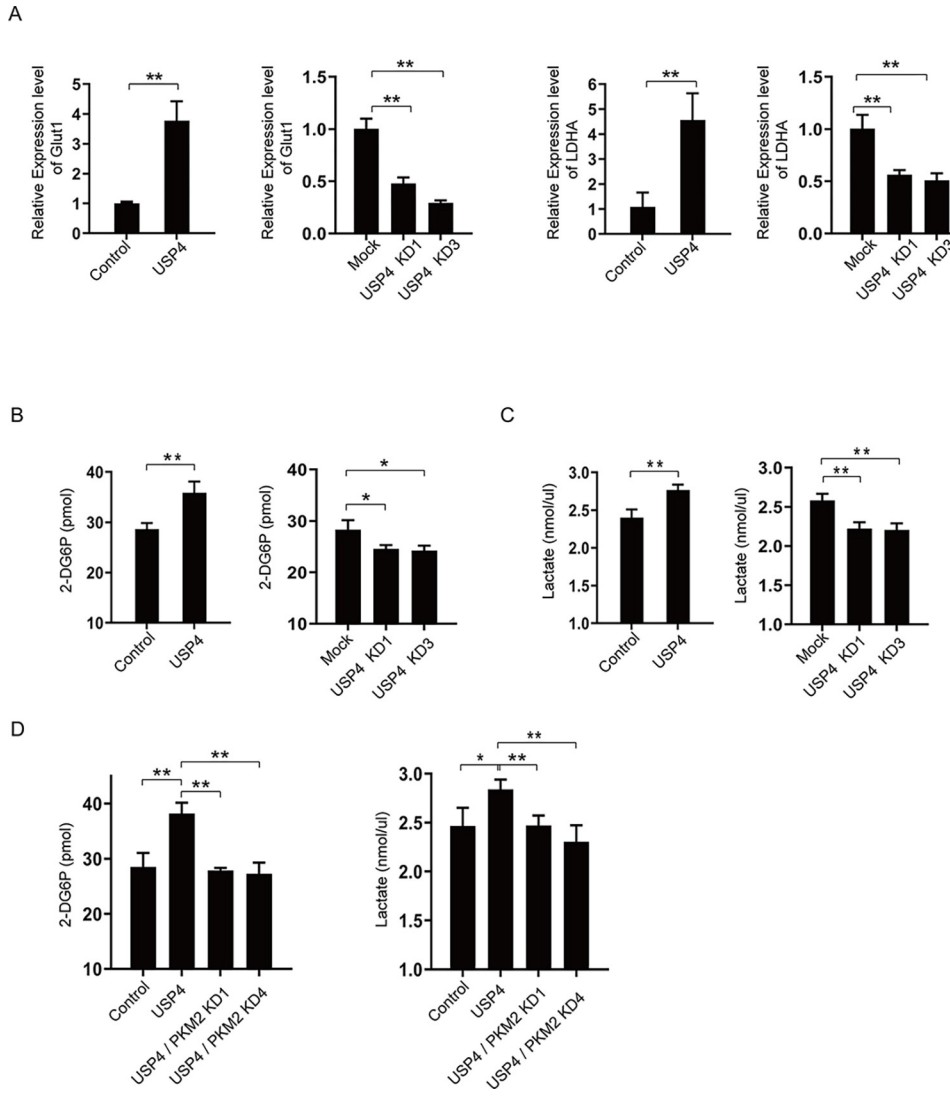

**Fig 4. USP4 enhances the Warburg effect of gastric cancer cells through its regulation of PKM2.** (A) The expression level of Glut1 and LDHA in USP4 overexpression and USP4 knockdown cells; (B) the glucose uptake of AGS cells when USP4 is overexpressed or knocked down; (C) the lactate production of AGS cells when USP4 is overexpressed or knocked down; (D) the glucose uptake and lactate production in AGS cells in which USP4 is overexpressed but PKM2 is knocked down.

The upregulated expression of the glycolytic genes, including PKM2, Glut1, and LDHA, by USP4 should enhance the Warburg effect. Thus, we examined the glucose uptake and lactate production in these cells and control cells. The higher level of USP4 caused higher glucose uptake in cells. When the expression level of USP4 was decreased by shRNAs, the glucose uptake of cells consequently went down (Fig 4B). As with the glucose uptake patterns, the production of lactate was changed with the expression level of USP4 (Fig 4C).

We have shown that the effect of USP4 on cell proliferation is mediated by PKM2. Is it the same situation for USP4's role in regulating glucose metabolism? To answer this, we used USP4 overexpression but knocked down PKM2 in cell lines for measuring glucose uptake and lactate production. There was no detectable expression difference in PKM1 between scrambled shRNA and PKM2 shRNA-treated cells (data not shown), suggesting that there was no

compensatory increase in PKM1 following knockdown of PKM2. We found that the stimulation of USP4 on glucose metabolism disappeared due to the PKM2 knockdown (Fig 4D). All the above results highlight the significance of USP4 in regulating the Warburg effect and indicate that its function is mediated by PKM2.

## Discussion

Ubiquitination modification is an important regulating mechanism for a number of cellular processes [34]. DUBs catalyze the reversible reaction of ubiquitination by removing ubiquitin from target proteins. Many studies have revealed the extensive regulatory functions of DUBs in tumor genesis and progression. Therefore, DUBs represent novel candidates for target-directed drug development [35, 36].

PKM2 is the rate-limiting enzyme of glycolysis and plays an important role in cancer metabolism and development. As an essential factor of the Warburg effect, PKM2 was found to be overexpressed in various cancers [7], and its expression level was often associated with the proliferation [37], migration/invasion [38, 39], malignancy [40], and lymphatic metastasis [41] of gastric cancer. The expression of PKM2 is regulated at multiple levels, including transcription, translation, and post-translation. Ubiquitination and deubiquitination are involved in regulating PKM2. Several DUBs have been found to regulate PKM2. For example, OTUB2 inhibits PKM2 ubiquitination and enhances its activity in colorectal cancer [42]. PMSD14 interacts with PKM2 and decreases PKM2 ubiquitination in ovarian cancer [43]. In addition, both USP20 [20] and HAUSP [44] can bind to PKM2 to regulate PKM2's ubiquitination.

The present study identified USP4 as a PKM2 interaction protein and found that USP4 can decrease the ubiquitination of PKM2. USP4 established its tumor-promoting function (enhancing cell proliferation and the Warburg effect) in gastric cancer cells by upregulating PKM2. This conclusion is supported by the following evidence: First, we confirmed the interaction between USP4 and PKM2. The results showed that USP4 is associated with PKM2, and they are co-localized in the cytosol and nucleus. The interaction region of USP4 with PKM2 is localized in its INSERT domain. However, further work is needed to exactly localize the interaction region of PKM2 with USP4. Second, USP4 increased the level of PKM2 in gastric cancer cells. In vitro ubiquitination assays demonstrated that USP4 catalyzed PKM2 deubiquitination. Third, USP4 increased cell proliferation and enhanced the Warburg effect through its regulation of PKM2.

Recent studies found that USP4 is often overexpressed in various cancers, such as glioblastoma [23], liver cancer [24, 25], colorectal cancer [26, 27], and melanoma [28], but its role in gastric cancer is unknown. Our current work indicates that USP4 not only promotes proliferation but also enhances the Warburg effect in gastric cancer cells by regulating PKM2 (Fig 5).

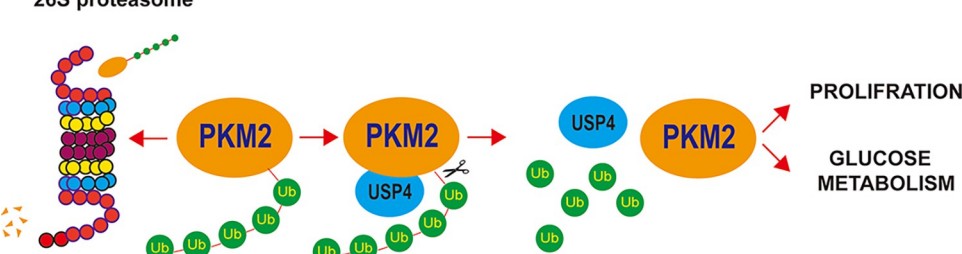

**Fig 5. The mechanism of USP4 regulating gastric cancer cell proliferation and glucose metabolism.** USP4 increases PKM2 stability via deubiquitination. Ub = ubiquitin.

Because of the limited condition of our animal facilities, we are not able to perform some in vivo experiments to further verify the tumor-promoting role of USP4 in gastric cancer development. However, based on our current data, we infer that USP4 may function as an oncogene in gastric cancer and could be a good candidate for target-directed gastric cancer drug development through further studies.

## Supporting information

**S1 Fig. Knockdown of PKM2 in USP4 overexpression cells.** Immunoblotting of USP4, PKM2, and GAPDH in AGS cells that are stably transfected with the USP4 expression plasmid and PKM2 knockdown plasmid, clones 1 and 4, respectively.
(TIF)

**S1 Raw images. All original images for blots and gels.**
(PDF)

**S1 Table. Primers for human USP4 ORF and deletion mutants.**
(PDF)

**S2 Table. Primers for human PKM2 ORF and deletion mutants.**
(PDF)

## Acknowledgments

We thank Dr. Qin Ye for her technical support.

## Author Contributions

**Conceptualization:** Dahu Chen.

**Data curation:** Yuanyuan Chen, Yunfei Guo, Mei Yuan, Song Guo, Shuaishuai Cui.

**Formal analysis:** Mei Yuan.

**Funding acquisition:** Dahu Chen.

**Investigation:** Yuanyuan Chen, Yunfei Guo, Mei Yuan, Song Guo.

**Methodology:** Mei Yuan.

**Supervision:** Dahu Chen.

**Writing – original draft:** Yuanyuan Chen, Dahu Chen.

**Writing – review & editing:** Dahu Chen.

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
