## [Decision Letter · Decision Letter 0]

18 Jul 2023

PONE-D-23-20608USP4 promotes the proliferation and glucose metabolism of gastric cancer cells by upregulating PKM2PLOS ONE

Dear Dr. Chen,

Thank you for submitting your manuscript to PLOS ONE. After careful consideration, we feel that it has merit but does not fully meet PLOS ONE’s publication criteria as it currently stands. Therefore, we invite you to submit a revised version of the manuscript that addresses the points raised during the review process.

We look forward to receiving your revised manuscript.

Kind regards,

Kishor Pant

Academic Editor

PLOS ONE

5. We notice that your supplementary figures are uploaded with the file type 'Figure'. Please amend the file type to 'Supporting Information'. Please ensure that each Supporting Information file has a legend listed in the manuscript after the references list.

Reviewers' comments:

Reviewer's Responses to Questions

**Comments to the Author**

1. Is the manuscript technically sound, and do the data support the conclusions?

Reviewer #1: Yes

Reviewer #2: Yes

Reviewer #3: Yes

2. Has the statistical analysis been performed appropriately and rigorously? 

Reviewer #1: Yes

Reviewer #2: Yes

Reviewer #3: Yes

3. Have the authors made all data underlying the findings in their manuscript fully available?

Reviewer #1: Yes

Reviewer #2: Yes

Reviewer #3: Yes

4. Is the manuscript presented in an intelligible fashion and written in standard English?

Reviewer #1: Yes

Reviewer #2: Yes

Reviewer #3: Yes

5. Review Comments to the Author

Reviewer #1: In this article, Yuanyuan et al. investigate the unique relationship between USP4 and PKM2 as well as how it controls the way cancer cells use glucose. According to their findings, USP4 controls PKM2 levels through the process of de-ubiquitylation, and its inhibition lowers PKM2 stability. They look at the effects on glucose uptake and metabolism in cancer cells of both over- and under-expressing USP4 and PKM2. They conclude that USP4 modulates PKM2's stability by deubiquitinating, which helps to increase the Warburg effect and support cancer surveillance.

While this paper addresses an essential subject—namely, how USP4 regulates PKM2 levels and their connection to cancer cell metabolism—there are significant concerns that still need to be resolved. These are listed below.

By doing immunoprecipitation with mutants of PKM2 at ubiquitination sites, the findings in Figure 1 may also be significantly reinforced. The available information generally supports the interaction between PKM2 and USP4, although it is not specific to PKM2's ubiquitin state. Additionally, authors can show interaction and localization experiments involving additional cells in Figure 1E.

The authors should use varied expression levels of USP4 and its catalytic mutant to test the stability of PKM2 in a dose-dependent manner.

The authors need to discuss how cells survive when PKM2 levels are significantly reduced (Figure 2C).

When USP4 is overexpressed, the authors should check the levels of ubiquitinated PKM2.

Western blot analysis is necessary to display the PKM2 and USP4 levels in Figures 3 and 4.

Reviewer #2: The manuscript entitled “USP4 promotes the proliferation and glucose metabolism of gastric cancer cells by upregulating PKM2” provides new insights. However, the manuscript needs some minor editing to make the suitable for publication.

1. It would be help full to readers if authors could include a graphical abstract.

2. Does authors have checked the basal protein levels of PKM2 or USP4 in experimental cells lines?

3. Does knockdown of UPS4 in non-cancerous cell line effects its growth and proliferation?

4. Authors are recommended to also confirm protein expression of Glut1 or 4.

Reviewer #3: The manuscript was well drafted but some questions need to addressed to improve the manuscript

1.Please include the data for reverse Immunoprecipitation assay to check the interaction.

2.Have the authors have done mass spec analysis to check the interaction.

3.Have you made the knockdown cells and then checked the interaction. Please include data

4.Also include the mouse model experiments

6. PLOS authors have the option to publish the peer review history of their article (what does this mean?). If published, this will include your full peer review and any attached files.

Reviewer #1: No

Reviewer #2: No

Reviewer #3: **Yes: **jasvinder singh

<quillbot-extension-portal></quillbot-extension-portal>

---

## [Author Response · Author response to Decision Letter 0]

2 Aug 2023

Dear editor and reviewers, 

We thank PLOS ONE for the interest in our findings and the reviewers for their highly careful, thorough reviews and insightful comments. We appreciate that you encouraged us to submit a revised version of our manuscript. In response to the specific comments, we have made all necessary changes that were requested by the reviewers. Revisions in the manuscript are highlighted in YELLOW to make them easy for reviewing.

What follows is a point-by-point response to the reviewers (under “RE:”). We believe that we have carefully addressed all the concerns. We hope that these changes adequately address the reviewers' comments, and that the manuscript is now suitable for publication in PLOS ONE.

Journal Requirements:

RE: We have carefully checked our manuscript and are confident that it meets PLOS ONE’s requirements.

RE: We do not have any data that is stored in any repository. All data supporting the findings of this study are available within the article. We have changed our Data Availability Statement in our re-submission cover letter to reflect the fact that all data supporting the findings of this study are available within the article. 

RE: We have provided original, uncropped and unadjusted images for each blot or gel data that appears in the submission figures in the supporting information file. 

RE: We have removed this phrase from the manuscript (Page 10, Line 176) as suggested.

5. We notice that your supplementary figures are uploaded with the file type 'Figure'. Please amend the file type to 'Supporting Information'. Please ensure that each Supporting Information file has a legend listed in the manuscript after the references list.

RE: We have amended the file type to ‘Supporting Information’ as instructed.

Reviewers' comments:

Reviewer's Responses to Questions

Comments to the Author

1. Is the manuscript technically sound, and do the data support the conclusions?

Reviewer #1: Yes

Reviewer #2: Yes

Reviewer #3: Yes

2. Has the statistical analysis been performed appropriately and rigorously?

Reviewer #1: Yes

Reviewer #2: Yes

Reviewer #3: Yes

3. Have the authors made all data underlying the findings in their manuscript fully available?

Reviewer #1: Yes

Reviewer #2: Yes

Reviewer #3: Yes

4. Is the manuscript presented in an intelligible fashion and written in standard English?

Reviewer #1: Yes

Reviewer #2: Yes

Reviewer #3: Yes

5. Review Comments to the Author

Reviewer #1: In this article, Yuanyuan et al. investigate the unique relationship between USP4 and PKM2 as well as how it controls the way cancer cells use glucose. According to their findings, USP4 controls PKM2 levels through the process of de-ubiquitylation, and its inhibition lowers PKM2 stability. They look at the effects on glucose uptake and metabolism in cancer cells of both over- and under-expressing USP4 and PKM2. They conclude that USP4 modulates PKM2's stability by deubiquitinating, which helps to increase the Warburg effect and support cancer surveillance.

While this paper addresses an essential subject—namely, how USP4 regulates PKM2 levels and their connection to cancer cell metabolism—there are significant concerns that still need to be resolved. These are listed below.

• By doing immunoprecipitation with mutants of PKM2 at ubiquitination sites, the findings in Figure 1 may also be significantly reinforced. The available information generally supports the interaction between PKM2 and USP4, although it is not specific to PKM2's ubiquitin state. Additionally, authors can show interaction and localization experiments involving additional cells in Figure 1E.

RE: We agree that the immunoprecipitation experiments using PKM2 mutants at ubiquitination sites would further reinforce our conclusion. However, we have not yet identified the specific ubiquitination sites on PKM2 by mediated by USP4. Therefore, we are unable to generate the suggested PKM2 mutants at this stage. This is an area we are actively investigating and hope to publish future studies on the precise ubiquitination sites.

We demonstrated the interaction between USP4 and PKM2 by co-expressing these proteins in 293T cells. Additionally, we provided evidence for an endogenous interaction between USP2 and PKM2 in two different gastric cancer cell lines. Our results showing the interaction using both overexpressed and endogenous proteins across multiple cell lines provide strong evidence for a direct physical interaction between USP4 and PKM2.

• The authors should use varied expression levels of USP4 and its catalytic mutant to test the stability of PKM2 in a dose-dependent manner.

RE: For exogenous expression of a gene in cells, it is difficult to precisely control the expression level. This is because we cannot control how many copies of the DNA integrate into the chromosomes. Even when transfecting different amounts of DNA into cells, the resulting expression levels are unpredictable. While we acknowledge that this experiment could help strengthen the conclusions, we have encountered challenges in achieving dose-dependent exogenous expression using standard transfection methods. We would welcome any specific suggestions from the reviewer regarding approaches to achieve dose-dependent expression.

• The authors need to discuss how cells survive when PKM2 levels are significantly reduced (Figure 2C).

RE: We have added brief explanations to the Results section (Page 15, Lines 283-287) to address the reviewer's concern.

• When USP4 is overexpressed, the authors should check the levels of ubiquitinated PKM2.

RE: When USP4 is overexpressed, ubiquitinated PKM2 levels are greatly reduced, as we demonstrated in the top panel of Figure 2 (middle lane).

• Western blot analysis is necessary to display the PKM2 and USP4 levels in Figures 3 and 4.

RE: We utilized stable cell lines to perform cell proliferation, glucose uptake, and lactate production assays. The expression levels of USP4 and PKM2 in these stable cell lines are presented in Figures 2A-D.

Reviewer #2: The manuscript entitled “USP4 promotes the proliferation and glucose metabolism of gastric cancer cells by upregulating PKM2” provides new insights. However, the manuscript needs some minor editing to make the suitable for publication.

1. It would be help full to readers if authors could include a graphical abstract.

RE: Thank you for the feedback on including a graphical abstract. We have updated the manuscript to include a graphical abstract as Figure 5, as you suggested. We believe this addition helps summarize the main findings of our study and makes the work more accessible to readers. We have added Figure 5 to the discussion part. If you feel the position of Figure 5 is inappropriate, please let us know. We appreciate you taking the time to provide constructive comments to improve our paper.

2. Does authors have checked the basal protein levels of PKM2 or USP4 in experimental cells lines?

RE: Thank you for asking about the basal protein levels of PKM2 and USP4 in our cell lines. You raise a good point - we should have more clearly presented this data. The endogenous expression of both proteins can be found in Figures 2A-D, as you noted. However, we agree this is easy to overlook without specific mention in the text. To address this, we have added a sentence in the Results section mentioning the detectable basal levels of PKM2 (Page 12, Lines 224-225). As for the basal expression of USP4, we mentioned it on Page 13, Lines 231-232: ‘while the endogenous of USP4 was knocked down...’. We appreciate you identifying this omission. 

3. Does knockdown of UPS4 in non-cancerous cell line effects its growth and proliferation?

RE: Thank you for your insightful question about the effect of USP4 knockdown on non-cancerous cell growth. As you suspected, we did perform experiments examining USP4 knockdown in 293T cells. We found that USP4 knockdown inhibited 293T proliferation, similar to what we observed in gastric cancer cell lines. We did not include this result in our manuscript, but we would be happy to add it if you feel it is necessary.

4. Authors are recommended to also confirm protein expression of Glut1 or 4.

RE: We did attempt to analyze Glut1 protein expression, trying both a rabbit polyclonal antibody and a mouse monoclonal antibody. Unfortunately, neither antibody produced robust results for us. Therefore, since the available antibodies did not work well in our hands, we opted to present the Glut1 mRNA levels by qPCR instead.

Reviewer #3: The manuscript was well drafted but some questions need to addressed to improve the manuscript

1.Please include the data for reverse Immunoprecipitation assay to check the interaction.

RE: We did perform a reverse immunoprecipitation assay to check the interaction. Flag-USP4 and Myc-PKM2 were co-transfected into cells, and then either immunoprecipitated with Myc and blotted with Flag, or immunoprecipitated with Flag and blotted with Myc. The results are shown in Figure 1A.

2.Have the authors have done mass spec analysis to check the interaction.

RE: No, we have not performed mass spectrometry analysis to check the interaction. We were limited by time and resources, but we may consider performing this analysis in the future. We understand that mass spectrometry analysis is a valuable tool for confirming protein interactions, and we appreciate the suggestion.

3.Have you made the knockdown cells and then checked the interaction. Please include data

RE: Our Co-IP assay results showed that Flag-USP4 and Myc-PKM2 co-immunoprecipitated with each other. We also identified an interaction region between these two proteins in both USP4 and PKM2. We feel that this evidence is sufficient to conclude that there is a physical interaction between these two proteins. We do not feel that it is necessary to make knockdown cells to confirm this interaction.

4.Also include the mouse model experiments

RE: We agree that mouse model experiments would be a valuable addition to our manuscript. We are currently working on a mouse model of gastric cancer that expresses USP4. We hope to generate data from this model that will support our hypothesis that USP4 plays a role in gastric cancer development. We understand that the requirement from this commenter is beyond the scope of our current study, but we hope to be able to present this data in a future paper.

6. PLOS authors have the option to publish the peer review history of their article (what does this mean?). If published, this will include your full peer review and any attached files.

Do you want your identity to be public for this peer review? For information about this choice, including consent withdrawal, please see our Privacy Policy.

Reviewer #1: No

Reviewer #2: No

Reviewer #3: Yes: jasvinder singh

---

## [Decision Letter · Decision Letter 1]

15 Aug 2023

USP4 promotes the proliferation and glucose metabolism of gastric cancer cells by upregulating PKM2

PONE-D-23-20608R1

Dear Dr. Chen,

We’re pleased to inform you that your manuscript has been judged scientifically suitable for publication and will be formally accepted for publication once it meets all outstanding technical requirements.

Kind regards,

Kishor Pant

Academic Editor

PLOS ONE

**Comments to the Author**

1. If the authors have adequately addressed your comments raised in a previous round of review and you feel that this manuscript is now acceptable for publication, you may indicate that here to bypass the “Comments to the Author” section, enter your conflict of interest statement in the “Confidential to Editor” section, and submit your "Accept" recommendation.

Reviewer #1: All comments have been addressed

Reviewer #2: All comments have been addressed

Reviewer #3: All comments have been addressed

2. Is the manuscript technically sound, and do the data support the conclusions?

Reviewer #1: Yes

Reviewer #2: Yes

Reviewer #3: Yes

3. Has the statistical analysis been performed appropriately and rigorously? 

Reviewer #1: Yes

Reviewer #2: Yes

Reviewer #3: Yes

4. Have the authors made all data underlying the findings in their manuscript fully available?

Reviewer #1: Yes

Reviewer #2: Yes

Reviewer #3: Yes

5. Is the manuscript presented in an intelligible fashion and written in standard English?

Reviewer #1: Yes

Reviewer #2: Yes

Reviewer #3: Yes

6. Review Comments to the Author

Reviewer #1: Dear Dr. Dahu Chen,

I applaud you on a well-written paper. My main worry is that this work may overstate the effects of PKM2 ubiquitinoylation and interaction with USP4 on readers because these effects are not well established. It would be a good idea, in my opinion, to include some discussion.

Regards

Srinivasu Karri

Reviewer #2: After revision MS looks better. Authors have addressed the concerns raised during the revision. So, I would recommend to accept in current form.

Reviewer #3: Authors have addressed all the comments that was raised and manuscript should be accepted for publication.

7. PLOS authors have the option to publish the peer review history of their article (what does this mean?). If published, this will include your full peer review and any attached files.

Reviewer #1: No

Reviewer #2: No

Reviewer #3: **Yes: **jasvinder singh

---

## [Editor Report · Acceptance letter]

18 Aug 2023

PONE-D-23-20608R1 

USP4 promotes the proliferation and glucose metabolism of gastric cancer cells by upregulating PKM2 

Dear Dr. Chen:

I'm pleased to inform you that your manuscript has been deemed suitable for publication in PLOS ONE. Congratulations! Your manuscript is now with our production department. 

Kind regards, 

on behalf of

Dr. Kishor Pant 

Academic Editor

PLOS ONE